# GABA_A_ Receptor Benzodiazepine Binding Sites and Motor Impairments in Parkinson’s Disease

**DOI:** 10.3390/brainsci13121711

**Published:** 2023-12-12

**Authors:** Nicolaas I. Bohnen, Jaimie Barr, Robert Vangel, Stiven Roytman, Rebecca Paalanen, Kirk A. Frey, Peter J. H. Scott, Prabesh Kanel

**Affiliations:** 1Department of Radiology, University of Michigan, Ann Arbor, MI 48109, USA; jaimieba@umich.edu (J.B.); rvangel@umich.edu (R.V.); stivenr@umich.edu (S.R.); kfrey@umich.edu (K.A.F.); pjhscott@umich.edu (P.J.H.S.); prabeshk@umich.edu (P.K.); 2Department of Neurology, University of Michigan, Ann Arbor, MI 48109, USA; rebecca.paalanen@gmail.com; 3Morris K. Udall Center of Excellence for Parkinson’s Disease Research, University of Michigan, Ann Arbor, MI 48109, USA; 4Neurology Service and GRECC, VA Ann Arbor Healthcare System, Ann Arbor, MI 48105, USA

**Keywords:** axial motor impairment, benzodiazepine binding site, dopamine, GABA_A_ receptor, Parkinson’s disease, PET

## Abstract

Flumazenil is an allosteric modulator of the γ-aminobutyric acid-A receptor (GABA_A_R) benzodiazepine binding site that could normalize neuronal signaling and improve motor impairments in Parkinson’s disease (PD). Little is known about how regional GABA_A_R availability affects motor symptoms. We investigated the relationship between regional availability of GABA_A_R benzodiazepine binding sites and motor impairments in PD. Methods: A total of 11 Patients with PD (males; mean age 69.0 ± 4.6 years; Hoehn and Yahr stages 2–3) underwent [^11^C]flumazenil GABA_A_R benzodiazepine binding site and [^11^C]dihydrotetrabenazine vesicular monoamine transporter type-2 (VMAT2) PET imaging and clinical assessment. Stepwise regression analysis was used to predict regional cerebral correlates of the four cardinal UPDRS motor scores using cortical, striatal, thalamic, and cerebellar flumazenil binding estimates. Thalamic GABA_A_R availability was selectively associated with axial motor scores (R^2^ = 0.55, F = 11.0, β = −6.4, *p* = 0.0009). Multi-ligand analysis demonstrated significant axial motor predictor effects by both thalamic GABA_A_R availability (R^2^ = 0.47, β = −5.2, F = 7.2, *p* = 0.028) and striatal VMAT2 binding (R^2^ = 0.30, β = −3.9, F = 9.1, *p* = 0.019; total model: R^2^ = 0.77, F = 11.9, *p* = 0.0056). Post hoc analysis demonstrated that thalamic [^11^C]methyl-4-piperidinyl propionate cholinesterase PET and *K*_1_ flow delivery findings were not significant confounders. Findings suggest that reduced thalamic GABA_A_R availability correlates with worsened axial motor impairments in PD, independent of nigrostriatal degeneration. These findings may augur novel non-dopaminergic approaches to treating axial motor impairments in PD.

## 1. Introduction

Axial motor impairments represent a significant cause of disability in Parkinson’s disease (PD). Dopaminergic medications are often not efficacious in treating these symptoms [1]. Cholinergic system dysfunction has been implicated in some components of postural instability and gait difficulties in PD, in particular falls and sensory processing during postural control, but not with overall severity of axial motor impairments when accounting for nigrostriatal nerve terminal losses [2,3,4]. Postural control and gait functions are mediated by widespread neural networks that cannot be captured by a simplistic model of single neurotransmitter system changes. There is increasing interest in the dysfunction of co-localized neurotransmitter functions to better understand the complexity of the multisystem nature of the neurodegeneration in PD.

γ-aminobutyric acid (GABA) is the major inhibitory neurotransmitter in the central nervous system. GABA binds to and mediates its effects via post-synaptic ionotropic GABA_A_ receptors (GABA_A_R) and pre- and post-synaptic metabotropic GABA_B_ receptors [5]. The role of GABA neurotransmission has been little studied in PD, despite the fact that the two major outflows of the basal ganglia, principal neurons of the globus pallidus internus, and of the substantia nigra pars reticulata largely employ inhibitory GABA to connect to areas outside the basal ganglia and that increased GABA activity from these nuclei has been demonstrated previously by both electrophysiologic and mRNA analyses in parkinsonian animal models [6,7,8]. Regional imbalance of the major inhibitory central nervous system transmitter activity may have propagating effects on the neuronal network activity underlying motor impairments in PD [9,10].

Benzodiazepine binding sites are present in a significant subset of cerebral GABA_A_ receptors [11]. Flumazenil is a short-acting intravenously administered silent allosteric modulator of the GABA_A_R benzodiazepine binding site, which rapidly improves motor impairments in PD, including postural instability and gait difficulties [12,13]. Based on current basal ganglia functional models of PD, flumazenil could affect neuronal signaling at several brain regions; there is, however, a knowledge gap about the relationship between availability of regional cerebral GABA_A_R benzodiazepine binding sites and specific motor impairments in human PD. There are relatively few in vivo imaging reports regarding the impairment of GABA_A_R benzodiazepine binding sites in the brain in PD. An in vivo imaging report by Japanese researchers found a correlation between reduced cerebral GABA_A_R benzodiazepine binding sites in the cortex as determined by [^123^I]iomazenil single-photon computed emission tomography and greater motor disability in PD but did not report on regions other than the cortex or striatal dopaminergic loss [14,15]. To address in more detail the role of GABA_A_R benzodiazepine binding sites, we investigated the relationship between in vivo regional cerebral availability of GABA_A_R benzodiazepine binding sites with [^11^C]flumazenil PET and motor impairments while accounting for nigrostriatal nerve terminal losses and cholinergic activity in subjects with PD.

## 2. Materials and Methods

### 2.1. Subjects and Clinical Test Battery

This cross-sectional study involved the analysis of 11 PD subjects (males, mean age 69.0 ± 4.6 years (SD; range 63–76); mean Mini-Mental State Examination score of 28.4 ± 2.4 (range 22–30); and mean duration of motor disease of 10.5 ± 4.1 years (range 5–15)). Subjects met the United Kingdom Parkinson’s Disease Society Brain Bank clinical diagnostic criteria [16]. Abnormal striatal [^11^C]dihydrotetrabenazine PET findings were consistent with the diagnosis of PD in all subjects. No subjects had a history of a large artery stroke or other significant intracranial disease. Mean modified Hoehn and Yahr stage was 2.6 ± 0.3 (range 2–3) with 1 subject in stage 2, 5 in stage 2.5 and 5 in stage 3 [17]. No subjects were taking benzodiazepine, (anti)cholinergic or neuroleptic drugs. Nine subjects were taking a combination of dopamine agonist and carbidopa–levodopa medications and two were using carbidopa–levodopa alone. All subjects completed the Unified Parkinson’s Disease Rating Scale (UPDRS) [18]. Subjects were examined and underwent [^11^C]dihydrotetrabenazine PET imaging in the morning after withholding dopaminergic drugs overnight. Mean motor UPDRS score was 28.3 ± 11.6 (range 10–48). UPDRS motor scores were divided into cardinal motor sub-scores for tremor (items 20 and 21), rigidity (item 22), distal appendicular bradykinesia (items 23–26 and 31), and axial symptoms (items 27–30). 

This study was approved by the Institutional Review Boards of Ann Arbor Department of Veterans Affairs Medical Center and the University of Michigan. Written informed consent was obtained from all subjects prior to any research procedures.

### 2.2. Imaging Techniques

All subjects underwent brain MRI, GABA_A_R benzodiazepine binding site imaging using [^11^C]flumazenil PET, and [^11^C]dihydrotetrabenazine vesicular monoamine transporter type-2 (VMAT2) PET imaging. Acetylcholinesterase PET imaging using the [^11^C]methyl-4-piperidinyl propionate (PMP) ligand was available in 10 subjects for additional analysis. [^11^C]flumazenil, [^11^C]dihydrotetrabenazine and [^11^C]PMP were prepared as described previously [19,20,21]. A bolus/infusion protocol was used for [^11^C]flumazenil dynamic PET imaging with intravenous bolus injection containing 40% of the total administered 10 mCi [^11^C]flumazenil dosage over 15 s, followed by continuous infusion of the remaining tracer at a constant rate for 62 min [19]. A bolus infusion was also used for [^11^C]dihydrotetrabenazine vesicular monoamine transporter type 2 (VMAT2) dynamic PET imaging with bolus injection of 55% of a 15 mCi dose, while the remaining 45% of the dose was continuously infused over the next 60 min [22]. Dynamic acetylcholinesterase PET scanning was performed for 70 min following an intravenous bolus of 15 mCi [^11^C]PMP. VMAT2 (used to quantify the degree of nigrostriatal striatal dopaminergic denervation) and acetylcholinesterase PET (used to quantify cholinergic thalamic binding) denervation were used for our post-analysis. The three PET scans were performed as part of a single study with two PET scans on the same day and the third one within days.

MRI was performed on a 3 Tesla Philips Achieva system (Philips, Best, The Netherlands) and PET imaging was performed in 3D imaging mode with an ECAT EXACT HR+ tomograph (Siemens Molecular Imaging, Inc., Knoxville, TN, USA) as previously reported [2].

### 2.3. Imaging Analysis

All image frames were spatially coregistered within subjects with a rigid-body transformation to reduce the effects of subject motion during the imaging session [23]. Interactive Data Language (IDL version 8.7) image analysis software (Research systems, Inc., Boulder, CO, USA) was used to manually trace volumes of interest on the MRI scan. Traced volumes of interest included the bilateral striatum (putamen and caudate nucleus), thalamus, pons, cerebellum, and neocortex. Neocortical volume of interest definition used semi-automated threshold delineation of the neocortical grey matter signal on the MRI images [24]. 

[^11^C]flumazenil distribution volume ratios were estimated using the Logan plot graphical analysis method [25]. The input kinetics for the reference tissue were derived from the pons, where the [^11^C]flumazenil binding is predominantly accounted for by free and nonspecifically bound radiotracer [26,27]. [^11^C]dihydrotetrabenazine distribution volume ratios were estimated also using the Logan plot graphical analysis method [25] with the striatal time activity curves as the input function and the total neocortex as reference tissue, a reference region overall low in VMAT2 binding sites, with the assumption that the non-displaceable distribution is uniform across the brain at equilibrium to allow accurate and stable assessment of VMAT2 binding when using the distribution volume ratio [22]. Acetylcholinesterase [^11^C]PMP hydrolysis rates (*k*_3_) were estimated using the striatal volume as the input tissue region [28]. 

### 2.4. Statistical Analysis

Stepwise regression analyses were used to predict cortical, striatal, thalamic and cerebellar flumazenil binding estimates from the four cardinal motor UPDRS scores as defined in Section 2.1. Analyses were performed using SAS version 9.3, (SAS institute, Cary, NC, USA). We also performed post hoc confounder analysis for the dopaminergic and cholinergic PET ligands. Post hoc confounder analysis was also performed using the *K*_1_ proxy flow images extracted from the flumazenil PET kinetic model. A model was considered significant if its *p*-value fell below our Bonferroni-adjusted α of 0.0125 (0.05/4 models).

## 3. Results

### 3.1. Availability of Regional Cerebral GABA_A_R Benzodiazepine Binding Sites and UPDRS Motor Scores

Significant findings were present for the thalamic region with axial motor scores as the only significant variable in the model (R^2^ = 0.55, F = 11.0, β = −6.4, *p* = 0.0009, significant after correction for multiple testing). Tremor was the only variable that entered the model for the cortex (R^2^ = 0.42, F = 6.6, β = −4.9, *p* = 0.03), which was no longer significant after correction for the effects of multiple testing. No cardinal UPDRS motor scores entered the regression models for the striatum or cerebellum.

### 3.2. Post Hoc Analysis of Thalamic GABA_A_R Benzodiazepine Binding Site Availability, Acetylcholinesterase Hydrolysis Rate, and VMAT2 and Axial UPDRS Motor Scores

A subgroup of 10 subjects completed all three PET ligand studies. Stepwise regression analysis was used to best predict axial UPDRS motor scores from thalamic GABA_A_R benzodiazepine binding site availability, thalamic acetylcholinesterase hydrolysis rate, striatal VMAT2 activity, age, and duration of motor disease. The overall model was significant (R^2^ = 0.77, F = 11.9, *p* = 0.0056) with significant contributions from both the thalamic GABA_A_R benzodiazepine binding site availability (R^2^ = 0.47, β = −5.2, F = 7.2, *p* = 0.028), and striatal VMAT2 binding (R^2^ = 0.30, β = −3.9, F = 9.1, *p* = 0.019). Thalamic acetylcholinesterase hydrolysis rates, age, and duration of disease did not meet the entry criteria for the model.

### 3.3. Post Hoc Analysis of Thalamic [^11^C]Flumazenil K_1_ Flow Effects and Axial UPDRS Motor Scores

Although the above findings show that other neurotransmitters, such as dopamine or acetylcholine were not confounders for our GABA_A_R findings, we used the *K*_1_ proxy flow images from the flumazenil PET kinetic model as an additional step to confirm that neural processes other than the two non-GABAergic neurotransmitters (dopamine and acetylcholine) may not play a significant role. This is because reduced gray matter flow is a marker of the global neurodegenerative process (or global neural integrity) and may be associated with glutamatergic activity (the most common neurotransmitter in the brain). For this purpose, we computed thalamic *K*_1_ flow derived from the [^11^C]flumazenil kinetic model. Results showed no significant effect of thalamic *K*_1_ flow measures in the prediction of axial motor UPDRS scores (F = 2.92, β = −2.6, *p* = 0.13). Furthermore, entering the thalamic *K*_1_ flow measure together with the thalamic [^11^C]flumazenil receptor binding measure not only failed to show a significant effect for the thalamic *K*_1_ flow measure but actually further strengthened the effect of the [^11^C]flumazenil receptor binding measure in the prediction of axial motor scores (F = 20.3, β = −6.7, *p* = 0.0028; total model F = 8.9, *p* = 0.009, Figure 1). 

## 4. Discussion

The thalamus is a key structure involved in motor control. The thalamus receives inhibitory inputs from the basal ganglia and excitatory signals from the cerebellum and cortex. Additional modulation in the form of monoaminergic and cholinergic signaling acts on the thalamus. Together, inputs to the thalamus act to modulate information received from the cortical regions resulting in motor control [29]. Our findings indicate that decreased availability of thalamic GABA_A_R benzodiazepine binding sites, reflecting increased GABAergic activity, is correlated with increased axial motor impairments in PD, independent of the degree of nigrostriatal degeneration. This may be compatible with the postulated basal ganglia model that the dopamine-denervated striatal nuclei provide inhibitory control over the globus pallidus internus and the substantia nigra pars reticulata [30], effectively “releasing” the tonic GABAergic inhibition mediated by the output structures of the basal ganglia [31]. As such, any dopaminergic hypoactivity within the striatum would therefore lead to a relative increase in inhibitory outflow from the basal ganglia [32]. Consequently, the subthalamic nucleus sends strong excitatory efferent signals to the globus pallidus internus and the substantia nigra pars reticulata, meaning that any increase in the firing rate of subthalamic nucleus neurons leads directly to an increased firing rate within globus pallidus internus and the substantia nigra pars reticulata neurons, in turn inhibiting the thalamic and brainstem structures resulting in mobility disturbances in PD [33]. In short, the dopaminergic hypoactivity in the striatum results in a comparative abundance of GABAergic inhibitory outputs from the basal ganglia to the thalamic region that ultimately leads to increased axial impairment in PD. 

Indeed, recent research supports this model; one study demonstrated a correlation between increased GABA in the basal ganglia and axial motor impairment in PD [34], and another study demonstrated that GABA_A_R antagonism restored dopaminergic firing in the striatum and improved motor symptoms in mouse models [35]. There is also evidence that direct thalamic pathology may contribute to the pathophysiology of motor impairments in PD. For example, a post-mortem study demonstrated a 30–50% loss of cells in the center-median/parafascicular complex, which normally provides important glutaminergic feedback from the thalamus to the putamen [29]. [^11^C]flumazenil binding site densities may serve as an indicator of synaptic neuropil integrity or may be an indicator of a disease-specific disturbance at the GABA_A_ receptor level. The former rationale is derived from the fact that GABA receptors are expressed on virtually all cortical and subcortical synaptic terminals. We performed a post hoc analysis to test the possibility that thalamic [^11^C]flumazenil GABA_A_R benzodiazepine binding site availability findings may be confounded by loss of neuronal integrity. For this purpose, we computed thalamic *K*_1_ flow measures derived from the [^11^C]flumazenil kinetic model but did not find a significant effect. Similar findings have been reported for the [^123^I]iomazenil single-photon computed emission tomography studies in PD where the authors also found an association between reduced cerebral GABA_A_R benzodiazepine binding site availability and increased motor disability in PD, which could not be explained by flow or perfusion data, suggesting a specific alteration of GABA_A_ receptors rather than of generalized synaptic or neuronal integrity [15].

These findings support our previous reports on in vivo imaging studies that demonstrated correlations between cholinergic innervation changes and posture, falls, and sensory processing during postural changes [3,24,36,37]. These findings agree with pharmacological studies, showing a benefit of cholinesterase drug treatment and a reduction in falls in subjects with PD with no significant changes in parkinsonian motor rating scores [38,39]. 

Our results also showed an independent effect for the integrity of nigrostriatal dopaminergic nerve terminals and axial motor impairments in PD. Although axial motor impairments are relatively refractory to dopaminergic treatments, a subset of these impairments are or remain responsive to these drugs [40]. Furthermore, there is emerging evidence for GABA and dopamine co-releasing neurotransmission from substantia nigra pars compacta (SNpc) and ventral tegmental area dopaminergic neurons [41,42]. Animal models of PD suggest that dopaminergic activity in the SNpc may be inhibited due to aberrant tonic inhibition, thought to be the result of excessive astrocytic GABA, leading to further imbalance between dopamine and GABA [43]. These observations illustrate the intricate interplay between these two major neurotransmitter changes and how it may be derailed by nigrostriatal denervation [8].

Our findings of an association between motor impairments and altered GABA_A_R availability may not be limited to Lewy body parkinsonism but may potentially also apply to other types of parkinsonian disorders. For example, a [^11^C]flumazenil GABA_A_R benzodiazepine binding site PET study in patients with vascular parkinsonism with and without gait disturbances found that striatal [^11^C]flumazenil uptake was inversely correlated with the motor UPDRS scores and [^11^C]flumazenil binding reductions were associated with the presence of gait disturbance [44]. However, comparisons of these findings and our present results are limited as—at least pure—vascular parkinsonism would not manifest with nigrostriatal degeneration [45], which, inherent to its dysfunction, would lead to a relative increase in inhibitory outflow from the basal ganglia [32].

As previously stated, axial symptoms of PD are often resistant to treatment with dopaminergic replacement therapy [1]. Additionally, treatment with deep brain stimulation (DBS), which commonly targets the subthalamic nucleus and globus pallidus internus, fails to alleviate axial symptoms and may worsen axial disability. Subthalamic nucleus DBS, specifically, has been correlated with greater axial impairment post-surgery [46]. This is in line with our findings as increased firing of excitatory efferents from the subthalamic nucleus may lead to increased inhibition of the thalamic and brainstem regions. Alternative treatments to address the relative hyper GABAergic activity in these regions may lead to breakthroughs in the treatment of axial impairment. 

Our findings augur GABA_A_R benzodiazepine binding site allosteric modulator drug treatment approaches to manage axial motor impairments in PD. Flumazenil is a fused imidazobenzodiazepine, which serves therapeutically as a GABA_A_R benzodiazepine binding site blocker [47]. Ondo and Hunter reported findings of single-dose (0.5 mg) intravenous flumazenil administration in eight PD patients and found significant improvements in total UPDRS motor scores, where the axial motor UPDRS sub-score tended to account for most of this improvement [12]. These flumazenil treatment data are compatible with the more selective association of reduced GABA_A_R benzodiazepine binding site availability and axial motor scores in our study.

Limitations of this study include the size and homogeneity of our sample. Because the patients were recruited from a veteran’s hospital, all the participants were male. This, in combination with the limited sample size, could influence the generalizability of the present findings with specific axial motor impairments, such as falls or freezing of gait. In addition, research into specific subtypes of axial impairments, such as patients with falls or freezing of gait, may lead to symptom-specific findings of GABA_A_R benzodiazepine binding site availability. Another limitation is the lack of a normal control or active disease control group to allow for the investigation of differential effects from normal aging or disease-specific effects of PD. Further studies based on a larger, more diverse study population, preferably with longitudinal follow up, are needed to more thoroughly investigate the role of GABA_A_R benzodiazepine binding site availability in the neural network underlying motor impairments in PD.

We conclude that thalamic GABA_A_R benzodiazepine binding site availability is inversely correlated with axial motor impairments in PD, independent from the degree of nigrostriatal degeneration. These findings may augur novel non-dopaminergic approaches to treating axial motor impairments in PD.

## Figures and Tables

**Figure 1 brainsci-13-01711-f001:**
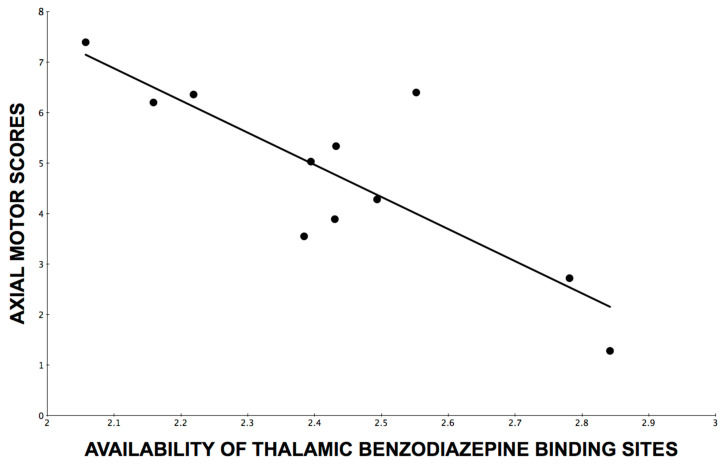
Line of best fit through a scatterplot of axial motor impairment scores over availability of thalamic benzodiazepine binding sites as assessed by [^11^C]flumazenil PET distribution volume ratios.

## Data Availability

The data that support the findings of this study are available on reasonable request from the corresponding author. The data are not publicly available due to their containing information that could compromise the privacy of research participants.

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
