# Peer review of "GABAA Receptor Benzodiazepine Binding Sites and Motor Impairments in Parkinson’s Disease"

_brainsci, 2023, doi:10.3390/brainsci13121711_

Round 1
Reviewer 1 Report
Comments and Suggestions for Authors
1. The study's findings are based on a small sample of only 11 patients. This limited sample size may affect the generalizability of the results and could lead to a higher risk of statistical errors. The patient population in this study is exclusively male, which limits the applicability of the findings to a broader, more diverse PD population. Explicitly address the limitations mentioned, such as the small sample size and the male-only patient population. Discuss how these limitations might impact the generalizability of the findings and suggest directions for future research to overcome these limitations. The conclusions drawn are cautious but might benefit from further discussion about the limitations of the study.
2. The study does not include a control group of healthy individuals or those with other neurological conditions. A control group would provide a baseline for comparison, aiding in the differentiation of PD-specific changes from those occurring in the general or diseased population.
3. In the manuscript you focus on the thalamus, thus it is important to briefly mention the role of the thalamus in axial motor impairments in PD, and explain the interaction between the GABAA receptors and dopamine.
4. The non-significant effect of thalamic K1 flow measures and the unexpected strengthening of the flumazenil receptor binding measure's impact need thorough clarification.
5. The study acknowledges that it did not separately analyze individuals with distinct types of axial motor impairments, these specific impairments could have different and unique associations with GABAAR benzodiazepine binding site.
6. The discussion could be enriched by a more comprehensive consideration of how these findings fit into the broader context of PD research. Discussing the implications of this research in terms of future therapeutic strategies or diagnostic approaches would provide valuable insights. Discuss how the observed correlation between thalamic GABAAR binding site availability and axial motor impairments could inform future treatment approaches. Connect the findings to potential therapeutic strategies, especially regarding GABAAR benzodiazepine binding site allosteric modulator drug treatment.
7. Further elaborate on the rationale behind the post hoc analysis involving thalamic [11C] flumazenil and K1 flow measures. Explain why this analysis was conducted and its relevance to the study's main findings. This will enhance the reader's understanding of how neuronal integrity considerations contribute to the interpretation of GABAAR binding site availability.
8. The discussion mentions dopaminergic hypoactivity within the striatum leading to increased inhibitory outflow from the basal ganglia. Elaborate on how this relates to the observed correlation between thalamic GABAAR binding site availability and axial motor impairments. Help readers understand the mechanistic connection between dopamine-denervated striatal nuclei and increased GABAergic activity in the context of motor impairments.
9. Mention how the assumption made in 2.3 Imaging analysis line 124 could affect the interpretation of data.
Reviewer 2 Report
Comments and Suggestions for Authors
The authors have described their findings in relation to [11C]flumazenil PET imaging changes in people with Parkinson's disease. The conclude, among others, that thalamic GABA-A receptor availability was reduced and selectively associated with axial motor scores. The findings are interesting and relatively novel but I have some concerns and suggestions for the manuscript.
1. Could they authors please clarify how the different PET scans (GABA-A, VMAT, and ACh) were obtained in the participant cohort. Were these done at the same point in time or were part of the presented data collected retrospectively? Moreover, were the VMAT and ACh PET scans part of the same research study for which ethical approval was obtained or were these collected as part of another studies/other studies?
2. In the statistical analysis section only [11C]flumazenil is mentioned but no statistical plan for the other PET ligands is provided.
3. Line 130 ''...from the four cardinal motor UPDRS scores''. I feel a definition of these scores should be provided along with a reference.
4. What was the rational for including an ACh PET target in this study?
5. Line 172 ''....independent of the degree of nigrostriatal degeneration''. Could the authors please clarify how they determine the degree of nigrostriatal degeneration? Where ioflupane DaT available for analysis in this patient cohort? If so, I would suggest showing this data and how they relate GABA-A receptor availability in an appropriate statistical model.
Round 2
Reviewer 1 Report
Comments and Suggestions for Authors
Thank you for taking all comments into consideration. The manuscript is better now.
Reviewer 2 Report
Comments and Suggestions for Authors
I would like to thank the authors for taking the time to amend their manuscript and address the comments that were raised. I have no further suggestions for improvement.